# Unsupervised Risk Estimation Using Only Conditional Independence Structure

**Jacob Steinhardt**
Stanford University
jsteinhardt@cs.stanford.edu

**Percy Liang**
Stanford University
pliang@cs.stanford.edu

## Abstract

We show how to estimate a model's test error from unlabeled data, on distributions very different from the training distribution, while assuming only that certain conditional independencies are preserved between train and test. We do not need to assume that the optimal predictor is the same between train and test, or that the true distribution lies in any parametric family. We can also efficiently compute gradients of the estimated error and hence perform unsupervised discriminative learning. Our technical tool is the method of moments, which allows us to exploit conditional independencies in the absence of a fully-specified model. Our framework encompasses a large family of losses including the log and exponential loss, and extends to structured output settings such as conditional random fields.

## 1 Introduction

Can we measure the accuracy of a model at test time without any ground truth labels, and without assuming the test distribution is close to the training distribution? This is the problem of *unsupervised risk estimation* (Donmez et al., 2010): Given a loss function $L(\theta; x, y)$ and a fixed model $\theta$, estimate the risk $R(\theta) \stackrel{\text{def}}{=} \mathbf{E}_{x,y \sim p^*}[L(\theta; x, y)]$ with respect to a test distribution $p^*(x, y)$, given access only to $m$ unlabeled examples $x^{(1:m)} \sim p^*(x)$. Unsupervised risk estimation lets us estimate model accuracy on a novel distribution, and is thus important for building reliable machine learning systems. Beyond evaluating a single model, it also provides a way of harnessing unlabeled data for learning: by minimizing the estimated risk over $\theta$, we can perform unsupervised learning and domain adaptation.

Unsupervised risk estimation is impossible without some assumptions on $p^*$, as otherwise $p^*(y \mid x)$—about which we have no observable information—could be arbitrary. How satisfied we should be with an estimator depends on how strong its underlying assumptions are. In this paper, we present an approach which rests on surprisingly weak assumptions—that $p^*$ satisfies certain conditional independencies, but not that it lies in any parametric family or is close to the training distribution.

To give a flavor for our results, suppose that $y \in \{1, \ldots, k\}$ and that the loss decomposes as a sum of three parts: $L(\theta; x, y) = \sum_{v=1}^{3} f_v(\theta; x_v, y)$, where the $x_v$ ($v = 1, 2, 3$) are independent conditioned on $y$. In this case, we show that we can estimate the risk to error $\epsilon$ in $\text{poly}(k)/\epsilon^2$ samples, independently of the dimension of $x$ or $\theta$, with only very mild additional assumptions on $p^*$. In Sections 2 and 3 we generalize to a larger family of losses including the log and exponential losses, and extend beyond the multiclass case to conditional random fields.

Some intuition behind our result is provided in Figure 1. At a fixed value of $x$, we can think of each $f_v$ as "predicting" that $y = j$ if $f_v(x_v, j)$ is low and $f_v(x_v, j')$ is high for $j' \neq j$. Since $f_1$, $f_2$, and $f_3$ all provide independent signals about $y$, their rate of agreement gives information about the model accuracy. If $f_1$, $f_2$, and $f_3$ all predict that $y = 1$, then it is likely that the true $y$ equals 1 and the loss is small. Conversely, if $f_1$, $f_2$, and $f_3$ all predict different values of $y$, then the loss is likely

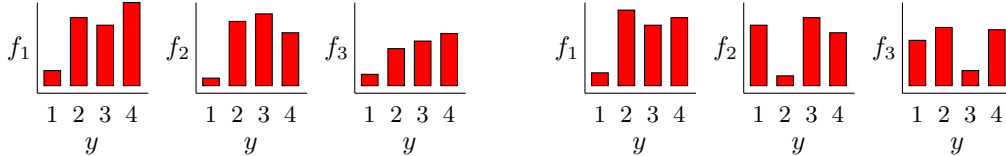

Figure 1: Two possible loss profiles at a given value of $x$. Left: if $f_1$, $f_2$, and $f_3$ are all minimized at the same value of $y$, that is likely to be the correct value and the total loss is likely to be small. Right: conversely, if $f_1$, $f_2$, and $f_3$ are small at differing values of $y$, then the loss is likely to be large.

large. This intuition is formalized by Dawid and Skene (1979) when the $f_v$ measure the $0/1$-loss of independent classifiers; in particular, if $r_v$ is the prediction of a classifier based on $x_v$, then Dawid and Skene model the $r_v$ as independent given $y$: $p(r_1, r_2, r_3) = \sum_{j=1}^{k} p(y = j) \prod_{v=1}^{3} p(r_v \mid y = j)$. They then use the learned parameters of this model to compute the $0/1$-loss.

**Partial specification.** Dawid and Skene's approach relies on the prediction $r_v$ only taking on $k$ values. In this case, the full distribution $p(r_1, r_2, r_3)$ can be parametrized by $k \times k$ conditional probability matrices $p(r_v \mid y)$ and marginals $p(y)$. However, as shown in Figure 1, we want to estimate continuous losses such as the log loss. We must therefore work with the prediction vector $f_v \in \mathbf{R}^k$ rather than a single predicted output $r_v \in \{1, \ldots, k\}$. To fully model $p(f_1, f_2, f_3)$ would require nonparametric estimation, resulting in an undesirable sample complexity exponential in $k$—in contrast to the discrete case, conditional independence effectively only *partially specifies* a model for the losses.

To sidestep this issue, we make use of the *method of moments*, which has recently been used to fit non-convex latent variable models (e.g. Anandkumar et al., 2012). In fact, it has a much older history in the econometrics literature, where it is used as a tool for making causal identifications under structural assumptions, even when no explicit form for the likelihood is known (Anderson and Rubin, 1949; 1950; Sargan, 1958; 1959; Hansen, 1982; Powell, 1994; Hansen, 2014). It is this latter perspective that we draw upon. The key insight is that even in the absence of a fully-specified model, certain moment equations–such as $\mathbf{E}[f_1 f_2 \mid y] = \mathbf{E}[f_1 \mid y]\mathbf{E}[f_2 \mid y]$–can be derived solely from the assumed conditional independence. Solving these equations yields estimates of $\mathbf{E}[f_v \mid y]$, which can in turn be used to estimate the risk. Importantly, our procedure avoids estimation of the full loss distribution $p(f_1, f_2, f_3)$, on which we make no assumptions other than conditional independence.

Our paper is structured as follows. In Section 2, we present our basic framework, and state and prove our main result on estimating the risk. In Section 3, we extend our framework in several directions, including to conditional random fields. In Section 4, we present a gradient-based learning algorithm and show that the sample complexity needed for learning is $d \cdot \text{poly}(k)/\epsilon^2$, where $d$ is the dimension of the parameters $\theta$. In Section 5, we investigate how our method performs empirically.

**Related Work.** While the formal problem of unsupervised risk estimation was only posed recently by Donmez et al. (2010), several older ideas from domain adaptation and semi-supervised learning are also relevant. The *covariate shift assumption* posits access to labeled samples from a training distribution $p_0(x, y)$ for which $p^*(y \mid x) = p_0(y \mid x)$. If $p^*(x)$ and $p_0(x)$ are close, we can approximate $p^*$ by $p_0$ via importance weighting (Shimodaira, 2000; Quiñonero-Candela et al., 2009). If $p^*$ and $p_0$ are not close, another approach is to assume a well-specified discriminative model family $\Theta$, such that $p_0(y \mid x) = p^*(y \mid x) = p_{\theta^*}(y \mid x)$ for some $\theta^* \in \Theta$; then the only error when moving from $p_0$ to $p^*$ is statistical error in the estimation of $\theta^*$ (Blitzer et al., 2011; Li et al., 2011). Such assumptions are restrictive—importance weighting only allows small perturbations from $p_0$ to $p^*$, and mis-specified models of $p(y \mid x)$ are common in practice; many authors report that mis-specification can lead to severe issues in semi-supervised settings (Merialdo, 1994; Nigam et al., 1998; Cozman and Cohen, 2006; Liang and Klein, 2008; Li and Zhou, 2015). More sophisticated approaches based on discrepancy minimization (Mansour et al., 2009) or learning invariant representations (Ben-David et al., 2006; Johansson et al., 2016) typically also make some form of the covariate shift assumption.

Our approach is closest to Dawid and Skene (1979) and some recent extensions (Zhang et al., 2014; Platanios, 2015; Jaffe et al., 2015; Fetaya et al., 2016). Similarly to Zhang et al. (2014) and Jaffe et al. (2015), we use the method of moments for estimating latent-variable models. However, those papers use it for parameter estimation in the face of non-convexity, rather than as a way to avoid full estimation of $p(f_v \mid y)$. The insight that the method of moments works under partial specification lets us extend beyond the simple discrete settings they consider to handle more complex continuous and structured losses. The intriguing work of Balasubramanian et al. (2011) provides an alternate approach

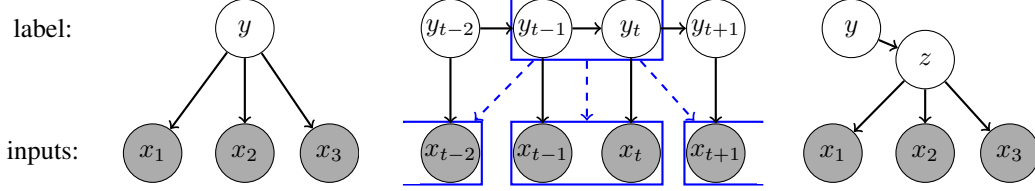

Figure 2: Left: our basic 3-view setup (Assumption 1). Center: Extension 1, to CRFs; the embedding of 3 views into the CRF is indicated in blue. Right: Extension 3, to include a mediating variable $z$.

to continuous losses; they show that the distribution of losses $L \,|\, y$ is often approximately Gaussian, and use that to estimate the risk. Among all this work, ours is the first to perform gradient-based learning and the first to handle a structured loss (the log loss for conditional random fields).

## 2 Framework and Estimation Algorithm

We will focus on multiclass classification; we assume an unknown true distribution $p^*(x, y)$ over $\mathcal{X} \times \mathcal{Y}$, where $\mathcal{Y} = \{1, \ldots, k\}$, and are given unlabeled samples $x^{(1)}, \ldots, x^{(m)}$ drawn i.i.d. from $p^*(x)$. Given parameters $\theta \in \mathbb{R}^d$ and a loss function $L(\theta; x, y)$, our goal is to estimate the risk of $\theta$ on $p^*$: $R(\theta) \stackrel{\text{def}}{=} \mathbf{E}_{x, y \sim p^*}[L(\theta; x, y)]$. Throughout, we will make the *3-view assumption*:

**Assumption 1** (3-view). *Under $p^*$, $x$ can be split into $x_1, x_2, x_3$, which are conditionally independent given $y$ (see Figure 2). Moreover, the loss decomposes additively across views: $L(\theta; x, y) = A(\theta; x) - \sum_{v=1}^{3} f_v(\theta; x_v, y)$, for some functions $A$ and $f_v$.*

Note that each $x_v$ can be large (e.g. they could be vectors in $\mathbf{R}^d$). If we have $V > 3$ views, we can combine views to obtain $V = 3$ without loss of generality. It also suffices for just the $f_v$ to be independent rather than the $x_v$. Given only 2 views, the risk can be shown to be unidentifiable in general, although obtaining upper bounds may be possible.

We give some examples where Assumption 1 holds, then state and prove our main result (see Section 3 for additional examples). We start with logistic regression, which will be our primary focus later on:

**Example 1** (Logistic Regression). Suppose that we have a log-linear model $p_\theta(y \mid x) = \exp\left(\theta^\top \left(\phi_1(x_1, y) + \phi_2(x_2, y) + \phi_3(x_3, y)\right) - A(\theta; x)\right)$, where $x_1$, $x_2$, and $x_3$ are independent conditioned on $y$. If our loss function is the log-loss $L(\theta; x, y) = -\log p_\theta(y \mid x)$, then Assumption 1 holds with $f_v(\theta; x_v, y) = \theta^\top \phi_v(x_v, y)$ and $A(\theta; x)$ equal to the partition function of $p_\theta$. □

Assumption 1 does *not* hold for the hinge loss (see Appendix A for details), but it does hold for a modified hinge loss, where we apply the hinge separately to each view:

**Example 2** (Modified Hinge Loss). Suppose that $L(\theta; x, y) = \sum_{v=1}^{3}(1 + \max_{j \neq y} \theta^\top \phi_v(x_v, j) - \theta^\top \phi_v(x_v, y))_+$. In other words, $L$ is the sum of 3 hinge losses, one for each view. Then Assumption 1 holds with $A = 0$, and $-f_v$ equal to the hinge loss for view $v$. □

The model can also be non-linear within each view $x_v$, as long as the views are combined additively:

**Example 3** (Neural Networks). Suppose that for each view $v$ we have a neural network whose output is a score for each of the $k$ classes, $(f_v(\theta; x_v, j))_{j=1}^{k}$. Sum the scores $f_1 + f_2 + f_3$, apply a soft-max, and evaluate using the log loss; then $L(\theta; x, y) = A(\theta; x) - \sum_{v=1}^{3} f_v(\theta; x_v, y)$, where $A(\theta; x)$ is the log-normalization constant of the softmax, and hence $L$ satisfies Assumption 1. □

We are now ready to present our main result on recovering the risk $R(\theta)$. The key starting point is the *conditional risk matrices* $M_v \in \mathbf{R}^{k \times k}$, defined as (suppressing the dependence on $\theta$)

$$(M_v)_{ij} = \mathbf{E}[f_v(\theta; x_v, i) \mid y = j]. \tag{1}$$

In the case of the 0/1-loss, the $M_v$ are confusion matrices; in general, $(M_v)_{ij}$ measures how strongly we predict class $i$ when the true class is $j$. If we could recover these matrices along with the marginal class probabilities $\pi_j \stackrel{\text{def}}{=} p^*(y = j)$, then estimating the risk would be straightforward; indeed,

$$R(\theta) = \mathbf{E}\left[A(\theta; x) - \sum_{v=1}^{3} f_v(\theta; x_v, y)\right] = \mathbf{E}[A(\theta; x)] - \sum_{j=1}^{k} \pi_j \sum_{v=1}^{3} (M_v)_{j,j}, \tag{2}$$

where $\mathbf{E}[A(\theta; x)]$ can be estimated from unlabeled data alone.

**Caveat: Class permutation.** Suppose that at training time, we learn to predict whether an image contains the digit $0$ or $1$. At test time, nothing changes except the definitions of $0$ and $1$ are reversed. It is clearly impossible to detect this from unlabeled data— mathematically, the risk matrices $M_v$ are only recoverable up to column permutation. We will end up computing the minimum risk over these permutations, which we call the *optimistic risk* and denote $\tilde{R}(\theta) \stackrel{\text{def}}{=} \min_{\sigma \in \text{Sym}(k)} \mathbf{E}_{x, y \sim p^*}[L(\theta; x, \sigma(y))]$. This equals the true risk as long as $\theta$ is at least aligned with the correct classes in the sense that $\mathbf{E}_x[L(\theta; x, j) \mid y = j] \leq \mathbf{E}_x[L(\theta; x, j') \mid y = j]$ for $j' \neq j$. The optimal $\sigma$ can be computed from $M_v$ and $\pi$ in $\mathcal{O}(k^3)$ time using maximum weight bipartite matching; see Section B for details.

Our main result, Theorem 1, says that we can recover both $M_v$ and $\pi$ up to permutation, with a number of samples that is polynomial in $k$:

**Theorem 1.** *Suppose Assumption 1 holds. Then, for any $\epsilon, \delta \in (0, 1)$, we can estimate $M_v$ and $\pi$ up to column permutation, to error $\epsilon$ (in Frobenius and $\infty$-norm respectively). Our algorithm requires*

$$m = \text{poly}\left(k, \pi_{\min}^{-1}, \lambda^{-1}, \tau\right) \cdot \frac{\log(2/\delta)}{\epsilon^2} \text{ samples to succeed with probability } 1 - \delta, \text{ where}$$

$$\pi_{\min} \stackrel{\text{def}}{=} \min_{j=1}^{k} p^*(y = j), \quad \tau \stackrel{\text{def}}{=} \mathbf{E}\left[\textstyle\sum_{v,j} f_v(\theta; x_v, j)^2\right], \quad \text{and } \lambda \stackrel{\text{def}}{=} \min_{v=1}^{3} \sigma_k(M_v), \tag{3}$$

*and $\sigma_k$ denotes the $k$th singular value. Moreover, the algorithm runs in time $m \cdot \text{poly}(k)$.*

Estimates for $M_v$ and $\pi$ imply an estimate for $\tilde{R}$ via (2); see Algorithm 1 below for details. Importantly, the sample complexity in Theorem 1 depends on the number of classes $k$, but not on the dimension $d$ of $\theta$. Moreover, Theorem 1 holds even if $p^*$ lies outside the model family $\theta$, and even if the train and test distributions are very different (in fact, the result is agnostic to how the model $\theta$ was produced). The only requirement is the 3-view assumption for $p^*$ and that $\lambda, \pi_{\min} \neq 0$.

Let us interpret each term in (3). First, $\tau$ tracks the variance of the loss, and we should expect the difficulty of estimating the risk to increase with this variance. The $\frac{\log(2/\delta)}{\epsilon^2}$ term is typical and shows up even when estimating the parameter of a random variable to accuracy $\epsilon$ from $m$ samples. The $\pi_{\min}^{-1}$ term appears because, if one of the classes is very rare, we need to wait a long time to observe even a single sample from that class, and even longer to estimate the risk on that class accurately.

Perhaps least intuitive is the $\lambda^{-1}$ term, which is large e.g. when two classes have similar conditional risk vectors $\mathbf{E}[(f_v(\theta; x_v, i))_{i=1}^{k} \mid y = j]$. To see why this matters, consider an extreme where $x_1, x_2$, and $x_3$ are independent not only of each other but also of $y$. Then $p^*(y)$ is completely unconstrained, and it is impossible to estimate $R$ at all. Why does this not contradict Theorem 1? The answer is that in this case, all rows of $M_v$ are equal and hence $M_v$ has rank 1, $\lambda = 0$, $\lambda^{-1} = \infty$, and we need infinitely many samples for Theorem 1 to hold; $\lambda$ measures how close we are to this degenerate case.

**Proof of Theorem 1.** We now outline a proof of Theorem 1. Recall the goal is to estimate the conditional risk matrices $M_v$, defined as $(M_v)_{ij} = \mathbf{E}[f_v(\theta; x_v, i) \mid y = j]$; from these we can recover the risk itself using (2). The key insight is that certain moments of $p^*(y \mid x)$ can be expressed as polynomial functions of the matrices $M_v$, and therefore we can solve for the $M_v$ even without explicitly estimating $p^*$. Our approach follows the technical machinery behind the spectral method of moments (e.g., Anandkumar et al., 2012), which we explain below for completeness.

Define the loss vector $h_v(x_v) = (f_v(\theta; x_v, i))_{i=1}^{k}$, which measures the loss that would be incurred under *each* of the $k$ classes. The conditional independence of the $x_v$ means that $\mathbf{E}[h_1(x_1)h_2(x_2)^\top \mid y] = \mathbf{E}[h_1(x_1) \mid y]\mathbf{E}[h_2(x_2) \mid y]^\top$, and similarly for higher-order conditional moments. Marginalizing over $y$, we see that there is low-rank structure in the moments of $h$ that we can exploit; in particular (letting $\otimes$ denote outer product and $A_{\cdot j}$ denote the $j$th column of $A$):

$$\mathbf{E}[h_v(x_v)] = \sum_{j=1}^{k} \pi_j \cdot (M_v)_{\cdot, j}, \quad \mathbf{E}[h_v(x_v) \otimes h_{v'}(x_{v'})] = \sum_{j=1}^{k} \pi_j \cdot (M_v)_{\cdot, j} \otimes (M_{v'})_{\cdot, j} \text{ for } v \neq v', \text{ and}$$

$$\mathbf{E}[h_1(x_1) \otimes h_2(x_2) \otimes h_3(x_3)] = \sum_{j=1}^{k} \pi_j \cdot (M_1)_{\cdot, j} \otimes (M_2)_{\cdot, j} \otimes (M_3)_{\cdot, j}. \tag{4}$$

The left-hand-side of each equation can be estimated from unlabeled data; using *tensor decomposition* (Lathauwer, 2006; Comon et al., 2009; Anandkumar et al., 2012; 2013; Kuleshov et al., 2015), it is

**Algorithm 1** Algorithm for estimating $\tilde{R}(\theta)$ from unlabeled data.

---
1: **Input**: unlabeled samples $x^{(1)}, \ldots, x^{(m)} \sim p^*(x)$.
2: Estimate the left-hand-side of each term in (4) using $x^{(1:m)}$.
3: Compute approximations $\hat{M}_v$ and $\hat{\pi}$ to $M_v$ and $\pi$ using tensor decomposition.
4: Compute $\sigma$ maximizing $\sum_{j=1}^k \hat{\pi}_{\sigma(j)} \sum_{v=1}^3 (\hat{M}_v)_{j,\sigma(j)}$ using maximum bipartite matching.
5: **Output**: estimated risk, $\frac{1}{m} \sum_{i=1}^m A(\theta; x^{(i)}) - \sum_{j=1}^k \hat{\pi}_{\sigma(j)} \sum_{v=1}^3 (\hat{M}_v)_{j,\sigma(j)}$.

---

then possible to solve for $M_v$ and $\pi$. In particular, we can recover $M$ and $\pi$ up to permutation: that is, we recover $\hat{M}$ and $\hat{\pi}$ such that $M_{i,j} \approx \hat{M}_{i,\sigma(j)}$ and $\pi_j \approx \hat{\pi}_{\sigma(j)}$ for some permutation $\sigma \in \mathrm{Sym}(k)$. This then yields Theorem 1; see Section C for a full proof.

Assumption 1 thus yields a set of moment equations (4) whose solution lets us estimate the risk without any labels $y$. The procedure is summarized in Algorithm 1: we (i) approximate the left-hand-side of each term in (4) by sample averages; (ii) use tensor decomposition to solve for $\pi$ and $M_v$; (iii) use maximum matching to compute the permutation $\sigma$; and (iv) use (2) to obtain $\tilde{R}$ from $\pi$ and $M_v$.

## 3 Extensions

Theorem 1 provides a basic building block which admits several extensions to more complex model structures. We go over several cases below, omitting most proofs to avoid tedium.

**Extension 1** (Conditional Random Field). Most importantly, the variable $y$ need not belong to a small discrete set; we can handle structured outputs such as a CRF as long as $p^*$ has the right structure. This contrasts with previous work on unsupervised risk estimation that was restricted to multiclass classification (though in a different vein, it is close to Proposition 8 of Anandkumar et al. (2012)).

Suppose that $p^*(x_{1:T}, y_{1:T})$ factorizes as a hidden Markov model, and that $p_\theta$ is a CRF respecting the HMM structure: $p_\theta(y_{1:T} \mid x_{1:T}) \propto \prod_{t=2}^T f_\theta(y_{t-1}, y_t) \cdot \prod_{t=1}^T g_\theta(y_t, x_t)$. For the log-loss $L(\theta; x, y) = -\log p_\theta(y_{1:T} \mid x_{1:T})$, we can exploit the decomposition

$$-\log p_\theta(y_{1:T} \mid x_{1:T}) = \sum_{t=2}^T \underbrace{-\log p_\theta(y_{t-1}, y_t \mid x_{1:T})}_{\overset{\mathrm{def}}{=} \ell_t} - \sum_{t=1}^T \underbrace{-\log p_\theta(y_t \mid x_{1:T})}_{\overset{\mathrm{def}}{=} \ell_t'}. \tag{5}$$

Each of the components $\ell_t$ and $\ell_t'$ satisfy Assumption 1 (see Figure 2; for $\ell_t$, the views are $x_{1:t-2}, x_{t-1:t}, x_{t+1:T}$, and for $\ell_t'$ they are $x_{1:t-1}, x_t, x_{t+1:T}$). We use Theorem 1 to estimate each $\mathbf{E}[\ell_t], \mathbf{E}[\ell_t']$ individually, and thus also the full risk $\mathbf{E}[L]$. (We actually estimate the risk for $y_{2:T-1} \mid x_{1:T}$ due to the 3-view assumption failing at the boundaries.)

In general, the idea in (5) applies to any structured output problem that is a sum of local 3-view structures. It would be interesting to extend our results to other structures such as more general graphical models (Chaganty and Liang, 2014) and parse trees (Hsu et al., 2012).

**Extension 2** (Exponential Loss). We can also relax the additivity $L = A - f_1 - f_2 - f_3$ in Assumption 1. For instance, suppose $L(\theta; x, y) = \exp(-\theta^\top \sum_{v=1}^3 \phi_v(x_v, y))$ is the exponential loss. Theorem 1 lets us estimate the matrices $M_v$ corresponding to $f_v(\theta; x_v, y) = \exp(-\theta^\top \phi_v(x_v, y))$. Then

$$R(\theta) = \mathbf{E}\left[\prod_{v=1}^3 f_v(\theta; x_v, y)\right] = \sum_j \pi_j \prod_{v=1}^3 \mathbf{E}\left[f_v(\theta; x_v, j) \mid y = j\right] \tag{6}$$

by conditional independence, so the risk can be computed as $\sum_j \pi_j \prod_{v=1}^3 (M_v)_{j,j}$. This idea extends to any loss expressible as $L(\theta; x, y) = A(\theta; x) + \sum_{i=1}^n \prod_{v=1}^3 f_i^v(\theta; x_v, y)$ for some functions $f_i^v$.

**Extension 3** (Mediating Variable). Assuming that $x_{1:3}$ are independent conditioned only on $y$ may not be realistic; there might be multiple subclasses of a class (e.g., multiple ways to write the digit 4) which would induce systematic correlations across views. To address this, we show that independence need only hold conditioned on a mediating variable $z$, rather than on the class $y$ itself.

Let $z$ be a refinement of $y$ (in the sense that knowing $z$ determines $y$) which takes on $k'$ values, and suppose that the views $x_1$, $x_2$, $x_3$ are independent conditioned on $z$, as in Figure 2. Then we can

try to estimate the risk by defining $L'(\theta; x, z) = L(\theta; x, y(z))$, which satisfies Assumption 1. The problem is that the corresponding risk matrices $M'_v$ will only have $k$ distinct rows and hence have rank $k < k'$. To fix this, suppose that the loss vector $h_v(x_v) = (f_v(x_v, j))_{j=1}^k$ can be extended to a vector $h'_v(x_v) \in \mathbf{R}^{k'}$, such that (i) the first $k$ coordinates of $h'_v(x_v)$ are $h_v(x_v)$ and (ii) the conditional risk matrix $M'_v$ corresponding to $h'_v$ has full rank. Then, Theorem 1 allows us to recover $M'_v$ and hence also $M_v$ (since it is a sub-matrix of $M'_v$) and thereby estimate the risk.

# 4 From Estimation to Learning

We now turn our attention to unsupervised learning, i.e., minimizing $R(\theta)$ over $\theta \in \mathbf{R}^d$. Unsupervised learning is impossible without some additional information, since even if we could learn the $k$ classes, we wouldn't know which class had which label (this is the same as the class permutation issue from before). Thus we assume that we have a small amount of information to break this symmetry:

**Assumption 2** (Seed Model). *We have access to a "seed model" $\theta_0$ such that $\tilde{R}(\theta_0) = R(\theta_0)$.*

Assumption 2 is very weak — it merely asks for $\theta_0$ to be aligned with the true classes on average. We can obtain $\theta_0$ from a small amount of labeled data (semi-supervised learning) or by training in a nearby domain (domain adaptation). We define $\mathrm{gap}(\theta_0)$ to be the difference between $R(\theta_0)$ and the next smallest permutation of the classes–i.e., $\mathrm{gap}(\theta_0) \stackrel{\mathrm{def}}{=} \min_{\sigma \neq \mathrm{id}} \mathbf{E}[L(\theta_0; x, \sigma(y)) - L(\theta_0; x, y)]$–which will affect the difficulty of learning.

For simplicity we will focus on the case of logistic regression, and show how to learn given only Assumptions 1 and 2. Our algorithm extends to general losses, as we show in Section F.

**Learning from moments.** Note that for logistic regression (Example 1), we have

$$R(\theta) = \mathbf{E}\Big[A(\theta; x) - \theta^\top \sum_{v=1}^3 \phi_v(x_v, y)\Big] = \mathbf{E}[A(\theta; x)] - \theta^\top \bar{\phi}, \text{ where } \bar{\phi} \stackrel{\mathrm{def}}{=} \sum_{v=1}^3 \mathbf{E}[\phi_v(x_v, y)]. \quad (7)$$

From (7), we see that it suffices to estimate $\bar{\phi}$, after which all terms on the right-hand-side of (7) are known. Given an approximation $\hat{\phi}$ to $\bar{\phi}$ (we will show how to obtain $\hat{\phi}$ below), we can learn a near-optimal $\theta$ by solving the following convex optimization problem:

$$\hat{\theta} = \underset{\|\theta\|_2 \leq \rho}{\arg\min} \mathbf{E}[A(\theta; x)] - \theta^\top \hat{\phi}. \quad (8)$$

In practice we would need to approximate $\mathbf{E}[A(\theta; x)]$ by samples, but we ignore this for simplicity (it generally only contributes lower-order terms to the error). The reason for the $\ell^2$-constraint on $\theta$ is that it imparts robustness to the error between $\hat{\phi}$ and $\bar{\phi}$. In particular (see Section D for a proof):

**Lemma 1.** *Suppose $\|\hat{\phi} - \bar{\phi}\|_2 \leq \epsilon$. Then the output $\hat{\theta}$ from (8) satisfies $R(\hat{\theta}) \leq \min_{\|\theta\|_2 \leq \rho} R(\theta) + 2\epsilon\rho$.*

If the optimal $\theta^*$ has $\ell^2$-norm at most $\rho$, Lemma 1 says that $\hat{\theta}$ nearly minimizes the risk: $R(\hat{\theta}) \leq R(\theta^*) + 2\epsilon\rho$. The problem of learning $\theta$ thus reduces to computing a good estimate $\hat{\phi}$ of $\bar{\phi}$.

**Computing $\hat{\phi}$.** Estimating $\bar{\phi}$ can be done in a manner similar to how we estimated $R(\theta)$ in Section 2. In addition to the conditional risk matrix $M_v \in \mathbf{R}^{k \times k}$, we compute the *conditional moment matrix* $G_v \in \mathbf{R}^{dk \times k}$, which tracks the conditional expectation of $\phi_v$: $(G_v)_{i+(r-1)k, j} \stackrel{\mathrm{def}}{=} \mathbf{E}[\phi_v(\theta; x_v, i)_r \mid y = j]$, where $r$ indexes $1, \ldots, d$. We then have $\bar{\phi}_r = \sum_{j=1}^k \pi_j \sum_{v=1}^3 (G_v)_{j+(r-1)k, j}$.

As in Theorem 1, we can solve for $G_1$, $G_2$, and $G_3$ using a tensor factorization similar to (4), though some care is needed to avoid explicitly forming the $(kd) \times (kd) \times (kd)$ tensor that would result (since $\mathcal{O}(k^3 d^3)$ memory is intractable for even moderate values of $d$). We take a standard approach based on random projections (Halko et al., 2011) and described in Section 6.1.2 of Anandkumar et al. (2013). We refer the reader to the aforementioned references for details, and cite only the resulting sample complexity and runtime, which are both roughly $d$ times larger than in Theorem 1.

**Theorem 2.** *Suppose that Assumptions 1 and 2 hold. Let $\delta < 1$ and $\epsilon < \min(1, \mathrm{gap}(\theta_0))$. Then, given $m = \mathrm{poly}\left(k, \pi_{\min}^{-1}, \lambda^{-1}, \tau\right) \cdot \frac{\log(2/\delta)}{\epsilon^2}$ samples, where $\lambda$ and $\tau$ are as defined in (3),*

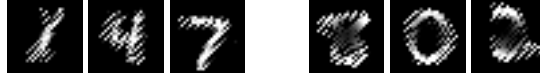

Figure 3: A few sample train images (left) and test images (right) from the modified MNIST data set.

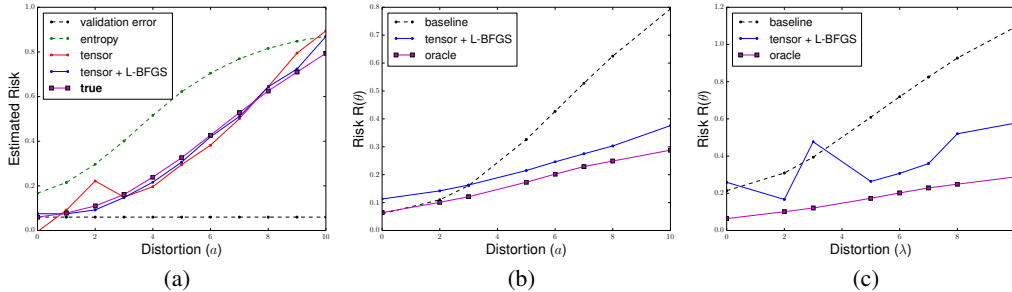

Figure 4: Results on the modified MNIST data set. (a) Risk estimation for varying degrees of distortion $a$. (b) Domain adaptation with 10,000 training and 10,000 test examples. (c) Domain adaptation with 300 training and 10,000 test examples.

*with probability $1 - \delta$ we can recover $M_v$ and $\pi$ to error $\epsilon$, and $G_v$ to error $(B/\tau)\epsilon$, where $B^2 = \mathbf{E}[\sum_{i,v} \|\phi_v(x_v, i)\|_2^2]$ measures the $\ell^2$-norm of the features. The algorithm runs in time $\mathcal{O}\left(d\left(m + \mathrm{poly}(k)\right)\right)$, and the errors are in Frobenius norm for $M$ and $G$, and $\infty$-norm for $\pi$.*

See Section E for a proof sketch. Whereas before we estimated the risk matrix $M_v$ to error $\epsilon$, now we estimate the gradient matrix $G_v$ (and hence $\bar{\phi}$) to error $(B/\tau)\epsilon$. To achieve error $\epsilon$ in estimating $G_v$ requires $(B/\tau)^2 \cdot \mathrm{poly}\left(k, \pi_{\min}^{-1}, \lambda^{-1}, \tau\right) \frac{\log(2/\delta)}{\epsilon^2}$ samples, which is $(B/\tau)^2$ times as large as in Theorem 1. The quantity $(B/\tau)^2$ typically grows as $\mathcal{O}(d)$, and so the sample complexity needed to estimate $\bar{\phi}$ is typically $d$ times larger than the sample complexity needed to estimate $R$. This matches the behavior of the supervised case where we need $d$ times as many samples for learning as compared to (supervised) risk estimation of a fixed model.

**Summary.** We have shown how to perform unsupervised logistic regression, given only a seed model $\theta_0$. This enables unsupervised learning under fairly weak assumptions (only the multi-view and seed model assumptions) even for mis-specified models and zero train-test overlap, and without assuming covariate shift. See Section F for learning under more general losses.

## 5  Experiments

To better understand the behavior of our algorithms, we perform experiments on a version of the MNIST data set that is modified to ensure that the 3-view assumption holds. To create an image $I$, we sample a class in $\{0, \ldots, 9\}$, then sample 3 images $I_1, I_2, I_3$ at random from that class, letting every third pixel in $I$ come from the respective image $I_v$. This guarantees there are 3 conditionally independent views. To explore train-test variation, we dim pixel $p$ in the image by $\exp\left(a\left(\|p - p_0\|_2 - 0.4\right)\right)$, where $p_0$ is the image center and distances are normalized to be at most 1. We show example images for $a = 0$ (train) and $a = 5$ (a possible test distribution) in Figure 3.

**Risk estimation.** We use Algorithm 1 to perform unsupervised risk estimation for a model trained on $a = 0$, testing on various values of $a \in [0, 10]$. We trained the model with AdaGrad (Duchi et al., 2010) on 10,000 training examples and 10,000 test examples to estimate the risk. To solve for $\pi$ and $M$ in (4), we first use the tensor power method implemented by Chaganty and Liang (2013) to initialize, and then locally minimize a weighted $\ell^2$-norm of the moment errors in (4) using L-BFGS. We compared with two other methods: (i) validation error from held-out samples (which would be valid if train = test), and (ii) predictive entropy $\sum_j -p_\theta(j \mid x) \log p_\theta(j \mid x)$ on the test set (which would be valid if the predictions were well-calibrated). The results are shown in Figure 4a; both the tensor method in isolation and tensor + L-BFGS estimate the risk accurately, with the latter performing slightly better.

**Unsupervised domain adaptation.** We next evaluate our learning algorithm in an unsupervised domain adaptation setting, where we receive labeled training data at $a = 0$ and unlabeled test data at a different value of $a$. We use the training data to obtain a seed model $\theta_0$, and then perform

unsupervised learning (Section 4), setting $\rho = 10$ in (8). The results are shown in Figure 4b. For small values of $a$, our algorithm performs worse than the baseline of directly using $\theta_0$, likely due to finite-sample effects. However, our algorithm is far more robust as $a$ increases, and tracks the performance of an oracle that was trained on the same distribution as the test examples.

Because we only need to provide our algorithm with a seed model for disentangling the classes, we do not need much data when training $\theta_0$. To verify this, we tried obtaining $\theta_0$ from only 300 labeled examples. Tensor decomposition sometimes led to bad initializations in this limited data regime, in which case we obtained a different $\theta_0$ by training with a smaller step size. The results are shown in Figure 4c. Our algorithm generally performs well, but has higher variability than before, seemingly due to higher condition number of the matrices $M_v$.

**Summary.** Our experiments show that given 3 views, we can estimate the risk and perform unsupervised domain adaptation, even with limited labeled data from the source domain.

## 6 Discussion

We have presented a method for estimating the risk from unlabeled data, which relies only on conditional independence structure and hence makes no parametric assumptions about the true distribution. Our approach applies to a large family of losses and extends beyond classification tasks to conditional random fields. We can also perform unsupervised learning given only a seed model that can distinguish between classes in expectation; the seed model can be trained on a related domain, on a small amount of labeled data, or any combination of the two, and thus provides a pleasingly general formulation highlighting the similarities between domain adaptation and semi-supervised learning.

Previous approaches to domain adaptation and semi-supervised learning have also exploited multi-view structure. Given two views, Blitzer et al. (2011) perform domain adaptation with zero source/target overlap (covariate shift is still assumed). Two-view approaches (e.g. co-training and CCA) are also used in semi-supervised learning (Blum and Mitchell, 1998; Ando and Zhang, 2007; Kakade and Foster, 2007; Balcan and Blum, 2010). These methods all assume some form of low noise or low regret, as do, e.g., transductive SVMs (Joachims, 1999). By focusing on the central problem of risk estimation, our work connects multi-view learning approaches for domain adaptation and semi-supervised learning, and removes covariate shift and low-noise/low-regret assumptions (though we make stronger independence assumptions, and specialize to discrete prediction tasks).

In addition to reliability and unsupervised learning, our work is motivated by the desire to build *machine learning systems with contracts*, a challenge recently posed by Bottou (2015); the goal is for machine learning systems to satisfy a well-defined input-output contract in analogy with software systems (Sculley et al., 2015). Theorem 1 provides the contract that under the 3-view assumption the test error is close to our estimate of the test error; the typical (weak) contract of ML systems is that if train and test are similar, then the test error is close to the training error. One other interesting contract is to provide prediction *regions* that contain the truth with probability $1 - \epsilon$ (Shafer and Vovk, 2008; Khani et al., 2016), which includes abstaining when uncertain as a special case (Li et al., 2011).

The most restrictive part of our framework is the three-view assumption, which is inappropriate if the views are not completely independent or if the data have structure that is not captured in terms of multiple views. Since Balasubramanian et al. (2011) obtain results under Gaussianity (which would be implied by many somewhat dependent views), we are optimistic that unsupervised risk estimation is possible for a wider family of structures. Along these lines, we end with the following questions:

**Open question.** In the 3-view setting, suppose the views are not completely independent. Is it still possible to estimate the risk? How does the degree of dependence affect the number of views needed?

**Open question.** Given only two independent views, can one obtain an upper bound on the risk $R(\theta)$?

The results of this paper have caused us to adopt the following perspective: To leverage unlabeled data, we should make *generative* structural assumptions, but still optimize *discriminative* model performance. This hybrid approach allows us to satisfy the traditional machine learning goal of predictive accuracy, while handling lack of supervision and under-specification in a principled way. Perhaps, then, what is truly needed for learning is understanding the *structure* of a domain.

**Acknowledgments.** This research was supported by a Fannie & John Hertz Foundation Fellowship, a NSF Graduate Research Fellowship, and a Future of Life Institute grant.

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
