[Supplementary Material 1 · full.pdf]

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

## A    Hinge Loss and Assumption 1

In Section 2 we stated that the hinge loss does not satisfy Assumption 1. In this section we explain why.

To be explicit, suppose that $x = (x_1, x_2, x_3)$, where the $x_v$ are independent conditioned on $y$, and let $L(\theta; x, y)$ be the multiclass hinge loss:

$$L(\theta; x, y) = \max_{j \neq y} \max \left( 1 + \theta^\top \sum_{v=1}^{3} (\phi(x_v, y) - \phi(x_v, j)), 0 \right). \tag{9}$$

To satisfy Assumption 1, $L(\theta; x, y)$ should decompose as

$$L(\theta; x, y) = A(\theta; x) - \sum_{v=1}^{3} f_v(\theta; x_v, y). \tag{10}$$

In particular, (10) implies that the dependence on $y$ should be additive over the views $v$. In (9), however, the $\max$ couples all of the views together, so that the decomposition (10) does not hold.

## B    Details of Computing $\tilde{R}$ from $M$ and $\pi$

In this section we show how, given $M$, and $\pi$, we can efficiently compute

$$\tilde{R}(\theta) = \mathbf{E}[A(\theta; x)] - \max_{\sigma \in \mathrm{Sym}(k)} \sum_{j=1}^{k} \pi_{\sigma(j)} \sum_{v=1}^{3} (M_v)_{j, \sigma(j)}. \tag{11}$$

The only bottleneck is the maximum over $\sigma \in \mathrm{Sym}(k)$, which would naïvely require considering $k!$ possibilities. However, we can instead cast this as a form of maximum matching. In particular, form the $k \times k$ matrix

$$X_{i,j} = \pi_i \sum_{v=1}^{3} (M_v)_{j,i}. \tag{12}$$

Then we are looking for the permutation $\sigma$ such that $\sum_{j=1}^{k} X_{\sigma(j),j}$ is maximized. If we consider each $X_{i,j}$ to be the weight of edge $(i, j)$ in a complete bipartite graph, then this is equivalent to asking for a matching of $i$ to $j$ with maximum weight, hence we can maximize over $\sigma$ using any maximum-weight matching algorithm such as the Hungarian algorithm, which runs in $\mathcal{O}\left(k^3\right)$ time (Tomizawa, 1971; Edmonds and Karp, 1972).

## C    Proof of Theorem 1

**Preliminary reductions.** Our goal is to estimate $M$ and $\pi$ to error $\epsilon$ (with probability of failure $1 - \delta$) in $\mathrm{poly}\left(k, \pi_{\min}^{-1}, \lambda^{-1}, \tau\right) \cdot \frac{\log(2/\delta)}{\epsilon^2}$ samples. Note that if we can estimate $M$ and $\pi$ to error $\epsilon$ with any fixed probability of success $1 - \delta_0 \geq \frac{3}{4}$, then we can amplify the probability of success to $1 - \delta$ at the cost of $\mathcal{O}(\log(2/\delta))$ times as many samples (the idea is to make several independent estimates, then throw out any estimate that is more than $2\epsilon$ away from at least half of the others; all the remaining estimates will then be within distance $3\epsilon$ of the truth with high probability).

**Estimating $M$.** Estimating $\pi$ and $M$ is mostly an exercise in interpreting Theorem 7 of Anandkumar et al. (2012), which we recall below, modifying the statement slightly to fit our language. Here $\kappa$ denotes condition number (which is the ratio of $\sigma_1(M)$ to $\sigma_k(M)$, since all matrices in question have $k$ columns).

**Theorem 3** (Anandkumar et al. (2012)). *Let $P_{v,v'} \overset{\text{def}}{=} \mathbf{E}[h_v(x) \otimes h_{v'}(x)]$, and $P_{1,2,3} \overset{\text{def}}{=} \mathbf{E}[h_1(x) \otimes h_2(x) \otimes h_3(x)]$. Also let $\hat{P}_{v,v'}$ and $\hat{P}_{1,2,3}$ be sample estimates of $P_{v,v'}$, $P_{1,2,3}$ that are (for technical convenience) estimated from independent samples of size $m$. Let $\|T\|_F$ denote the $\ell^2$-norm of $T$ after unrolling $T$ to a vector (e.g., when $T$ is a matrix $\|T\|_F$ is the Frobenius norm). Suppose that, for some constants $C_{1,2}$, $C_{1,3}$, $C_{1,2,3}$, we have:*

- $\mathbf{P}\left[\|\hat{P}_{v,v'} - P_{v,v'}\|_2 \leq C_{v,v'}\sqrt{\frac{1}{\delta m}}\right] \geq 1 - \delta$ *for* $\{v, v'\} \in \{\{1,2\}, \{1,3\}\}$, *and*

- $\mathbf{P}\left[\|\hat{P}_{1,2,3} - P_{1,2,3}\|_F \leq C_{1,2,3}\sqrt{\frac{1}{\delta m}}\right] \geq 1 - \delta.$

*Then, there exist universal constants* $C$, $m_0$, $\delta_0$ *such that the following holds: if* $m \geq m_0$ *and* $\delta \leq \delta_0$ *and*

$$\sqrt{\frac{k}{\delta m}} \leq C \cdot \frac{\min_{j \neq j'} \|(M_3^\top)_j - (M_3^\top)_{j'}\|_2 \cdot \sigma_k(P_{1,2})}{C_{1,2,3} \cdot k^5 \cdot \kappa(M_1)^4} \cdot \frac{\delta}{\log(k/\delta)} \cdot \epsilon$$

$$\sqrt{\frac{1}{\delta m}} \leq C \cdot \min\left\{\frac{\min_{j \neq j'} \|(M_3^\top)_j - (M_3^\top)_{j'}\|_2 \cdot \sigma_k(P_{1,2})^2}{C_{1,2} \cdot \|P_{1,2,3}\|_F \cdot k^5 \cdot \kappa(M_1)^4} \cdot \frac{\delta}{\log(k/\delta)}, \frac{\sigma_k(P_{1,3})}{C_{1,3}}\right\} \cdot \epsilon$$

*for some* $\epsilon < 1$, *then with probability at least* $1 - 5\delta$, *we can output* $\hat{M}_3$ *with the following guarantee: there exists a permutation* $\sigma \in \mathrm{Sym}(k)$ *such that for all* $j \in \{1, \ldots, k\}$,

$$\|(M_3^\top)_j - (\hat{M}_3^\top)_{\sigma(j)}\|_2 \leq \max_{j'} \|(M_3^\top)_{j'}\|_2 \cdot \epsilon. \tag{13}$$

By symmetry, we can use Theorem 3 to recover each of the matrices $M_v$, $v = 1, 2, 3$, up to permutation of the columns. Furthermore, Anandkumar et al. (2012) show in Appendix B.4 of their paper how to match up the columns of the different $M_v$, so that only a single unknown permutation is applied to each of the $M_v$ simultaneously. We will set $\delta = 1/180$, which yields a probability of success of $1 - \frac{5}{180}$ for recovering each individual $M_v$, and hence an overall probability of success of $1 - \frac{3 \cdot 5}{180} = 11/12$ for this part of the proof.

We now analyze the rate of convergence implied by Theorem 3. Note that by Chebyshev's inequality we can take $C_{1,2,3} = \mathcal{O}\left(\sqrt{\mathbf{E}[\|h_1\|_2^2\|h_2\|_2^2\|h_3\|_2^2]}\right)$, and similarly $C_{v,v'} = \mathcal{O}\left(\sqrt{\mathbf{E}[\|h_v\|_2^2\|h_{v'}\|_2^2]}\right)$. Next, observe that Theorem 3 implies that we can estimate the $M_v$ to error $\epsilon$ given $Z/\epsilon^2$ samples, where $Z$ is polynomial in the following quantities:

1. $k$,

2. $\max_{v=1}^3 \kappa(M_v)$, where $\kappa$ denotes condition number,

3. $\frac{\sqrt{\mathbf{E}[\|h_1\|_2^2\|h_2\|_2^2\|h_3\|_2^2]}}{\left(\min_{j,j'} \|(M_v^\top)_j - (M_v^\top)_{j'}\|_2\right) \cdot \sigma_k(P_{v',v''})}$, where $(v, v', v'')$ is a permutation of $(1, 2, 3)$,

4. $\frac{\|P_{1,2,3}\|_2}{\left(\min_{j,j'} \|(M_v^\top)_j - (M_v^\top)_{j'}\|_2\right) \cdot \sigma_k(P_{v',v''})}$, where $(v, v', v'')$ is as before, and

5. $\frac{\sqrt{\mathbf{E}[\|h_v\|_2^2\|h_{v'}\|_2^2]}}{\sigma_k(P_{v,v'})}$.

6. $\max_{j,v} \|(M_v^\top)_j\|_2$.

It suffices to show that each of these quantities are polynomial in $k$, $\pi_{\min}^{-1}$, $\tau$, and $\lambda^{-1}$.

(1) $k$ is trivially polynomial in itself.

(2) Note that $\kappa(M_v) \leq \sigma_1(M_v)/\lambda \leq \|M_v\|_F/\lambda$. Furthermore, $\|M_v\|_F^2 = \sum_j \|\mathbf{E}[h_v \mid j]\|_2^2 \leq \sum_j \mathbf{E}[\|h_v\|_2^2 \mid j] \leq k\tau^2$. In all, $\kappa(M_v) \leq \sqrt{k}\tau/\lambda$, which is polynomial in $k$ and $\tau/\lambda$.

(3) We first note that $\min_{j \neq j'} \|(M_v^\top)_j - (M_v^\top)_{j'}\|_2 = \sqrt{2}\min_{j \neq j'} \|M_v^\top(e_j - e_{j'})\|_2/\|e_j - e_{j'}\|_2 \geq \sqrt{2}\sigma_k(M_v)$. Also, $\sigma_k(P_{v',v''}) = \sigma_k(M_{v'}\mathrm{diag}(\pi)M_{v''}) \geq \sigma_k(M_{v'})\pi_{\min}\sigma_k(M_{v''})$. We can thus upper-bound the quantity in (3.) by

$$\frac{\sqrt{\mathbf{E}[\|h_1\|_2^2\|h_2\|_2^2\|h_3\|_2^2]}}{\sqrt{2}\pi_{\min}\sigma_k(M_1)\sigma_k(M_2)\sigma_k(M_3)} \leq \frac{\tau^3}{\sqrt{2}\pi_{\min}\lambda^3},$$

which is polynomial in $\pi_{\min}^{-1}$, $\tau/\lambda$.

(4) We can perform the same calculations as in (3), but now we have to bound $\|P_{1,2,3}\|_2$. However, it is easy to see that

$$
\begin{aligned}
\|P_{1,2,3}\|_2 &= \sqrt{\|\mathbf{E}[h_1 \otimes h_2 \otimes h_3]\|_2^2} \\
&\leq \sqrt{\mathbf{E}[\|h_1 \otimes h_2 \otimes h_3\|_2^2]} \\
&= \sqrt{\mathbf{E}[\|h_1\|_2^2 \|h_2\|_2^2 \|h_3\|_2^2]} \\
&= \sqrt{\sum_{j=1}^{k} \pi_j \prod_{v=1}^{3} \mathbf{E}[\|h_v\|_2^2 \mid y = j]} \\
&\leq \tau^3,
\end{aligned}
$$

which yields the same upper bound as in (3).

(5) We can again perform the same calculations as in (3), where we now only have to deal with a subset of the variables; we end up obtaining a bound of $\frac{\tau^2}{\pi_{\min}\lambda^2}$.

(6) We have $\|(M_v^\top)_j\|_2 = \|\mathbf{E}[h_v \mid y = j]\|_2 \leq \sqrt{\mathbf{E}[\|h_v\|_2^2 \mid y = j]} \leq \tau$.

In sum, we have shown that with probability $\frac{11}{12}$ we can estimate each $M_v$ to column-wise $\ell^2$ error $\epsilon$ using $\mathrm{poly}\left(k, \pi_{\min}^{-1}, \lambda^{-1}, \tau\right)/\epsilon^2$ samples; since there are only $k$ columns, we can make the total (Frobenius) error be at most $\epsilon$ while still using $\mathrm{poly}\left(k, \pi_{\min}^{-1}, \lambda^{-1}, \tau\right)/\epsilon^2$ samples. It now remains to estimate $\pi$.

**Estimating $\pi$.** This part of the argument follows Appendix B.5 of Anandkumar et al. (2012). Noting that $\pi = M_1^{-1}\mathbf{E}[h_1]$, we can estimate $\pi$ as $\hat{\pi} = \hat{M_1}^{-1}\hat{\mathbf{E}}[h_1]$, where $\hat{\mathbf{E}}$ denotes the empirical expectation. Hence, we have

$$
\|\pi - \hat{\pi}\|_\infty \leq \left\| (\hat{M_1}^{-1} - M_1^{-1})\mathbf{E}[h_1] + M_1^{-1}(\hat{\mathbf{E}}[h_1] - \mathbf{E}[h_1]) + (\hat{M_1}^{-1} - M_1^{-1})(\hat{\mathbf{E}}[h_1] - \mathbf{E}[h_1]) \right\|_\infty
$$

$$
\leq \underbrace{\|\hat{M_1}^{-1} - M_1^{-1}\|_F}_{(i)} \underbrace{\|\mathbf{E}[h_1]\|_2}_{(ii)} + \underbrace{\|M_1^{-1}\|_F}_{(iii)} \underbrace{\|\hat{\mathbf{E}}[h_1] - \mathbf{E}[h_1]\|_2}_{(iv)} + \underbrace{\|\hat{M_1}^{-1} - M_1^{-1}\|_F}_{(i)} \underbrace{\|\hat{\mathbf{E}}[h_1] - \mathbf{E}[h_1]\|_2}_{(iv)}.
$$

We will bound each of these factors in turn:

(i) $\|\hat{M_1}^{-1} - M_1^{-1}\|_F$: let $E_1 = \hat{M_1} - M_1$, which by the previous part satisfies $\|E_1\|_F \leq \sqrt{k}\max_j \|(\hat{M_1}^\top)_j - (M_1^\top)_j\|_2 = \mathrm{poly}\left(k, \pi_{\min}^{-1}, \lambda^{-1}, \tau\right)/\sqrt{m}$. Therefore:

$$
\begin{aligned}
\|\hat{M_1}^{-1} - M_1^{-1}\|_F &\leq \|(M_1 + E_1)^{-1} - M_1^{-1}\|_F \\
&= \|M_1^{-1}(I + E_1 M_1^{-1})^{-1} - M_1^{-1}\|_F \\
&\leq \|M_1^{-1}\|_F \cdot \sigma_1\left((I + E_1 M_1^{-1})^{-1} - I\right) \\
&\leq k\lambda^{-1} \cdot \sigma_1\left((I + E_1 M_1^{-1})^{-1} - I\right) \\
&\leq k\lambda^{-1} \frac{\sigma_1(E_1 M_1^{-1})}{1 - \sigma_1(E_1 M_1^{-1})} \\
&\leq k\lambda^{-2} \frac{\|E_1\|_F}{1 - \lambda^{-1}\|E_1\|_F} \\
&\leq \frac{\mathrm{poly}\left(k, \pi_{\min}^{-1}, \lambda^{-1}, \tau\right)}{1 - \mathrm{poly}\left(k, \pi_{\min}^{-1}, \lambda^{-1}, \tau\right)/\sqrt{m}} \cdot \frac{1}{\sqrt{m}}.
\end{aligned}
$$

We can assume that $m \geq \mathrm{poly}\left(k, \pi_{\min}^{-1}, \lambda^{-1}, \tau\right)$ without loss of generality (since otherwise we can trivially obtain the desired bound on $\|\pi - \hat{\pi}\|_\infty$ by simply guessing the uniform distribution), in which case the above quantity is $\mathrm{poly}\left(k, \pi_{\min}^{-1}, \lambda^{-1}, \tau\right) \cdot \frac{1}{\sqrt{m}}$.

(ii) $\|\mathbf{E}[h_1]\|_2$: as before, we have $\|\mathbf{E}[h_1]\|_2 \leq \sqrt{\mathbf{E}[\|h_1\|_2^2]} \leq \tau$.

(iii) $\|M_1^{-1}\|_F$: since $M_1$ has $k$ columns, we have $\|M_1^{-1}\|_F \le \sqrt{k}\sigma_1\left(M_1^{-1}\right) \le \sqrt{k}\lambda^{-1}$.

(iv) $\|\hat{\mathbf{E}}[h_1] - \mathbf{E}[h_1]\|_2$: with any fixed probability (in this case, $11/12$), this term is $\mathcal{O}\left(\sqrt{\frac{\mathbf{E}[\|h_1\|_2^2]}{m}}\right) = \mathcal{O}\left(\frac{\tau}{\sqrt{m}}\right)$ by Chebyshev's inequality.

In sum, with probability at least $\frac{11}{12}$ all of the terms are poly $\left(k, \pi_{\min}^{-1}, \lambda^{-1}, \tau\right)$, and at least one factor in each term has a $\frac{1}{\sqrt{m}}$ decay. Therefore, we have $\|\pi - \hat{\pi}\|_\infty \le$ poly $\left(k, \pi_{\min}^{-1}, \lambda^{-1}, \tau\right) \cdot \sqrt{\frac{1}{m}}$.

Since we have shown that we can estimate each of $M$ and $\pi$ individually with probability $\frac{11}{12}$, we can estimate them jointly with probability $\frac{5}{6} > \frac{3}{4}$, thus completing the proof.

## D  Proof of Lemma 1

Let $B(\rho) = \{\theta \mid \|\theta\|_2 \le \rho\}$. First note that $|\theta^\top(\hat{\phi} - \bar{\phi})| \le \|\theta\|_2\|\hat{\phi} - \bar{\phi}\|_2 \le \epsilon\rho$ for all $\theta \in B(\rho)$. Letting $\tilde{\theta}$ denote the minimizer of $R(\theta)$ over $B(\rho)$, we obtain

$$R(\hat{\theta}) = \mathbf{E}[A(\hat{\theta}; x)] - \hat{\theta}^\top\bar{\phi} \tag{14}$$

$$\le \mathbf{E}[A(\hat{\theta}; x)] - \hat{\theta}^\top\hat{\phi} + \epsilon\rho \tag{15}$$

$$\overset{(i)}{\le} \mathbf{E}[A(\tilde{\theta}; x)] - \tilde{\theta}^\top\hat{\phi} + \epsilon\rho \tag{16}$$

$$\le \mathbf{E}[A(\tilde{\theta}; x)] - \tilde{\theta}^\top\bar{\phi} + 2\epsilon\rho \tag{17}$$

$$= R(\tilde{\theta}) + 2\epsilon\rho, \tag{18}$$

as claimed, where (i) is because $\hat{\theta}$ is the minimizer of $\mathbf{E}[A(\theta; x)] - \theta^\top\hat{\phi}$, and the remaining inequalities follow from the observation above that $|\theta^\top(\hat{\phi} - \bar{\phi})| \le \epsilon\rho$.

## E  Proof of Theorem 2

We provide here a sketch of the proof of Theorem 2, which essentially follows from the proof of Theorem 1. We note that Theorem 7 of Anandkumar et al. (2012) (and hence Theorem 1 above) does not require that the $M_v$ be $k \times k$, but only that they have $k$ *columns* (the number of rows can be arbitrary). It thus applies for any matrix $M_v'$ whose $j$th columns is obtained as $\mathbf{E}[h_v'(x_v) \mid j]$ for some $h_v' : \mathcal{X}_v \to \mathbf{R}^{d'}$. In our specific case, we will take $h_v' : \mathcal{X}_v \to \mathbf{R}^{k(d+1)}$, where the first $k$ coordinates of $h'(x_v)$ are equal to $h(x_v)$ (i.e., $(f_v(x_v, i))_{i=1}^k$), and the remaining $kd$ coordinates of $h'(x_v)$ are equal to $\frac{\tau}{B}\phi_v(x_v, i)_r$ as in the definition of $G_v$, where the difference is that we have scaled by a factor of $\frac{\tau}{B}$. Note that in this case $M_v' = \begin{bmatrix} M_v \\ \frac{\tau}{B}G_v \end{bmatrix}$. We let $\lambda'$ and $\tau'$ denote the values of $\lambda$ and $\tau$ for $M'$ and $h'$.

Since $M_v$ is a submatrix of $M_v'$, we have $\sigma_k(M_v') \ge \sigma_k(M_v)$, so $\lambda' \ge \lambda$. For $\tau'$, note that

$$\tau' = \mathbf{E}\left[\sum_v \|h_v'(x_v)\|_2^2\right] \tag{19}$$

$$= \mathbf{E}\left[\sum_v \|h_v(x_v)\|_2^2 + \frac{\tau^2}{B^2}\sum_{v,i}\|\phi_v(x_v, i)\|_2^2\right] \tag{20}$$

$$= \tau^2 + \frac{\tau^2}{B^2}\mathbf{E}\left[\sum_{v,i}\|\phi_v(x_v, i)\|_2^2\right] \tag{21}$$

$$= 2\tau^2, \tag{22}$$

so $\tau' \le \sqrt{2}\tau$. Since $(\lambda')^{-1} = \mathcal{O}(\lambda^{-1})$ and $\tau' = \mathcal{O}(\tau)$, we still obtain a sample complexity of poly $\left(k, \pi_{\min}^{-1}, \lambda^{-1}, \tau\right) \cdot \frac{\log(2/\delta)}{\epsilon^2}$. Since $\theta_0$ satisfies Assumption 2, we can recover the correct

permutation of the columns of $M_v$ (and hence also of $G_v$, since they are permuted in the same way), which completes the proof.

## F   Learning with General Losses

In Section 4, we formed the conditional moment matrix $G_v$, which stored the conditional expectation $\mathbf{E}[\phi_v(x_v, i) \mid y = j]$ for each $j$ and $i$. However, there was nothing special about computing $\phi$ (as opposed to some other moments), and for general losses we can form the conditional gradient matrix $G_v(\theta)$, defined by

$$G_v(\theta)_{i+kr,j} = \mathbf{E}\left[\frac{\partial}{\partial \theta_r} f_v(\theta; x_v, i) \mid y = j\right]. \tag{23}$$

Theorem 2 applies identically to the matrix $G_v(\theta)$ at any fixed $\theta$. We can then compute the gradient $\nabla_\theta R(\theta)$ using the relationship

$$\frac{\partial}{\partial \theta_r} R(\theta) = \mathbf{E}\left[\frac{\partial}{\partial \theta_r} A(\theta; x)\right] - \sum_{j=1}^{k} \pi_j \sum_{v=1}^{3} G_v(\theta)_{j+kr,j}. \tag{24}$$

For clarity, we also use $M_v(\theta)$ to denote the conditional risk matrix at a value $\theta$. To compute the gradient $\nabla_\theta R(\theta)$, we jointly estimate $M_v(\theta_0)$ and $G_v(\theta)$ (note the differing arguments of $\theta_0$ vs. $\theta$). Since the seed model assumption (Assumption 2) allows us to recover the correct column permutation for $M_v(\theta_0)$, estimating $G_v(\theta)$ jointly with $M_v(\theta_0)$ ensures that we recover the correct column permutation for $G_v(\theta)$ as well.

The final ingredient in learning $\theta$ is any gradient descent procedure that is robust to errors in the gradient (so that after $T$ steps with error $\epsilon$ on each step, the total error is $\mathcal{O}(\epsilon)$ and not $\mathcal{O}(\epsilon T)$). Fortunately, this is the case for many gradient descent algorithms, including any algorithm that can be expressed as mirror descent (we omit the details because they are somewhat beyond our scope, but refer the reader to Lemma 21 of Steinhardt et al. (2016) for a proof of this in the case of exponentiated gradient).

The general learning algorithm is given in Algorithm 2:

---
**Algorithm 2** General algorithm for learning via gradient descent.

---
1: Parameters: step size $\eta$
2: Input: unlabeled samples $x^{(1)}, \ldots, x^{(m)} \sim p^*(x)$, seed model $\theta_0$
3: $z^{(1)} \leftarrow 0 \in \mathbf{R}^d$
4: **for** $t = 1$ **to** $T$ **do**
5:     $\theta^{(t)} \leftarrow \arg\min_\theta \frac{1}{2\eta}\|\theta - \theta_0\|_2^2 - \theta^\top z^{(t)}$
6:     Compute $(M_v^{(t)}, G_v^{(t)})$ by jointly estimating $M_v(\theta_0), G_v(\theta^{(t)})$ from $x^{(1:m)}$.
7:     **for** $r = 1$ **to** $d$ **do**
8:         $g_r \leftarrow \frac{1}{m} \sum_{i=1}^{m} \frac{\partial}{\partial \theta_r} A(\theta^{(t)}; x^{(i)}) - \sum_{j=1}^{k} \pi_j \sum_{v=1}^{3} (G_v^{(t)})_{j+kr,j}$
9:         $z_r^{(t+1)} \leftarrow z_r^{(t)} + g_r$
10:     **end for**
11: **end for**
12: Output $\frac{1}{T}\left(\theta^{(1)} + \cdots + \theta^{(T)}\right)$.

---

[Supplementary Material 2 · paper.pdf]