[Reviews · NeurIPS 2016]

Reviewer 1

Summary

The paper proposes a means to estimate the risk of a model on unlabeled data, and to optimise a model on the same. The approach relies on making a structural assumption of the underlying generating distribution, namely that it decomposes into label independent views. Preliminary experimental results show promise for the method.

Qualitative Assessment

I found the paper very well presented and enjoyable to read. The basic problem is interesting, and the approach presented as some salient features, notably the fact that one does not have to make parametric assumption on the underlying distribution. The high-level idea of imposing structural assumptions but nonetheless relying on discriminative models was quite elegant. The basic insight in estimating the risk from unlabelled data is that by encoding a certain structural assumption - namely, that the data comprises three independent views - one implicitly gets information about the class-conditional risks by considering the first three moments of the label vectors. This leads to a system of equations which may be solved to infer the class-conditional risks. One applies a similar trick in order to estimate the gradient of the risk from unlabelled data, allowing one to optimise models such as logistic regression without labels. The fact that one is able to do so under somewhat mild assumptions is quite surprising. The mechanics of the proposed approach, as well as the overall inspiration for considering higher-order moments, derive from the tensor decomposition framework for latent variable models, e.g. Anandkumar et al. 2012, a literature that I am not familiar with. I am thus unable to confidently proclaim the novelty of the presented approach. However, from some preliminary reading, it does not appear as though existing work in the literature has use such methods for unsupervised risk estimation. It seems a nice conceptual insight that one can use the three view assumption to aid in estimating classification error. The technical content is fairly clearly presented in the main body, with some high-level intuition for the proofs of the main theorems. The proofs of Theorem 1 and 2 to rely on Theorem 7 of Anandkumar et al. 2012, but I wasn't quite able to discern how much of the heavy lifting is accomplished by the latter. I also do not have much intuition on how good the precise form of the sample complexity is, in terms of dependence on various parameters, but the paper makes an attempt to explain why each of these is intuitively necessary. I view the contribution of the paper as primarily theoretical, but appreciate that some preliminary experiments are presented. They indicate that the proposed method can be useful in covariate shift scenarios. The performance for semi-supervised learning is a little more erratic, but there are likely many practical extensions possible. Overall, I found this an interesting, well-written paper that solves a non-trivial problem. Given that I am not an expert in the field, I cannot however specifically assess the novelty of the work. Minor comments: - the assumption that the last decomposes as a sum over the three views is intuitive. In the specific example of logistic regression, it was a little unclear why the feature maps for the three views have the same dimensionality. In general does one simply partition the ambient feature space three separate components for the view? - I didn't quite follow the statement that assumption one doesn't hold for the hinge loss. Which specific version of multiclass hinge are you referring to? - in equation 1 and elsewhere, maybe it would be good to make explicit that M depends on theta. - for the extension to hidden Markov models, it might be worth adding a line explaining the difference to what is done in e.g. Anandkumar et al. 2012, which also considers estimating parameters for such models. - the seed model in assumption 2 doesn't appear to be explicitly referred to beyond its definition in the body of the paper (i.e. it isn't said in the body what one actually does with it), which could be confusing. - isn't the domain adaptation experiment more specifically one of covariate shift? - it wasn't clear why the proposed model performed worse than the seed for small values of a in the domain adaptation experiments. Is this a finite sample effect? - citation for Anandkumar et al. 2013 could be for the JMLR version. - some citations in the main body only appear in the complete list of references in the supplementary. I suggest perhaps reducing spacing or font size for the references in the body to accommodate everything.

Confidence in this Review

1-Less confident (might not have understood significant parts)


Reviewer 2

Summary

The problem is estimating the risk of a classifier or a regression function using unlabeled data only. This original problem has been proposed in the prior work (Donmez, 2010). In the present paper the main assumption is that there are at least three "views" of each object, and these views are conditionally independent given the labels. Under some additional assumptions, certain performance guarantees are derived.

Qualitative Assessment

My main concern is that the main contribution/improvement with respect to (Donmez, 2010) is not made really clear. In particular, (Donmez, 2010) also introduces conditional independence (section 2.2 Collaborative Estimation of the Risks: Conditionally Independent Predictors) where the conditional independence is that of predictors rather than "views." In the paper, predictors also are decomposed view-wise, thus their predictions are also conditionally independent. Thus, the assumption appears stronger than the corresponding assumption from (Donmez, 2010). Is cond. independence of views (rather than predictors' outputs) actually needed? In any case, what is the precise improvement w.r.t. (Donmez, 2010)? Perhaps the authors can clarify this in the rebuttal. I admit that the problem is rather unusual so I may be not understanding something. (The fact that the problem is unusual I consider rather an advantage.) The HMM setting is somewhat misleading: it requires m i.i.d. time series, in order for the i.i.d. assumption to hold. Usually in HMMs one tries to make inference based on just one time series. UPDATE: The novelty aspect, which was my main criticism, is more clear to me now (mainly from reading the rest of the reviews), so I lift this objection.

Confidence in this Review

1-Less confident (might not have understood significant parts)


Reviewer 3

Summary

This paper presents a method to estimate the test error of a model from unlabeled data, on distributions that are different from the distribution of the training set. The approach is based on the method of moments and the assumption that the feature space is divided in 3 conditionally independent subspaces conditionally to the output (3-views assumption); but no assumption is made on the optimal predictor or the parametric form of the distribution. The method extends to structured output settings such as HMMs.

Qualitative Assessment

The paper is well written and I did not find any flaws in the proofs. My main concern is about the 3-views assumption and how realistic it could be? The assumption is stronger than the one of co-training which was proposed for multi-view learning and in some papers the assumption is discussed to be unrealistic for mono-view learning. I would suggest to have a discussion on whenever the main assumption can be relaxed or a least a discussion about how to deal the problem where it does not hold in many general cases.

Confidence in this Review

2-Confident (read it all; understood it all reasonably well)


Reviewer 4

Summary

Review of 1820 The unsupervised risk estimation approach is based on two key assumptions: The feature vector x can be partitioned as x=(x_1,x_2,…x_v) such that i) the class-conditional distributions factor into a product of distributions (one factor for each x_i) ii) the loss is a separable function (one term for each x_i) The authors state that (i) is a “surprisingly weak assumption,” but it seems rather strong to me. For example, if the class-conditional distributions are Gaussian then (i) requires that the covariances are block-diagonal. Perhaps the authors could better motivate the assumption by discussing it in relation to multi-view learning (the term “view” is not mentioned until page 3). Also, it might help to include a motivational real-world example in the introduction. Also (ii) seems to rule out most commonly used loss functions, such as the usually hinge and logistic losses, which are nonlinear and therefore non-separable in x. To illustrate, in Example 1 in the paper, the authors assume a logistic model of the form theta^T (phi_1(x_1,y)+phi_2(x_2,y)+phi_3(x_3,y)). The standard GLM logistic model (for a linear classifier) has the form y*w^T x = y*(w_1^T x_1+w_2^T x_2+w_3^T x_3), which can not be represented in the form of Example 1 unless the phi_i functions include learnable parameters. Example 2 in the paper proposes a separable form of hinge loss. This accommodates their approach, but it is a different loss than the usual hinge loss. Again, perhaps these sorts of modifications could be motivated in the context of multi-view learning. Past work in this area is said to handle only 0/1-loss. The main contribution of the paper is to consider the above framework (factorized class-conditional distributions and separable loss functions) with continuous losses. Since the losses are continuous, rather than counts, the authors propose a method of moments approach to keep the computations tractable. Specifically, under the “3-view” assumption, the approach is based on a finding a decomposition of a third order tensor.

Qualitative Assessment

The paper is fairly well written and the application of tensor decomposition methods to this sort of unsupervised risk estimation appears to be novel (although the tensor-based machinery used to prove the main theorem is pretty standard now). The paper would benefit from better motivation of the 3-view assumptions (conditional independence of views and separability of loss), which might be helped with a nice real-world application example in the introduction.

Confidence in this Review

2-Confident (read it all; understood it all reasonably well)


Reviewer 5

Summary

This paper provides a tensor-decomposition-based methodology for unsupervised risk estimation based on 3-view assumption. The result is then used to develop a semi-supervised learning algorithm.

Qualitative Assessment

This paper attempts to tackle and interesting and important problem of risk estimation in the absence of labeled data. I think some ideas are interesting (thoug perhaps not very novel ;see below). Overall, at this point this paper is borderline and decision will be based on the rebuttal and discussion. Question/comments: - The idea is based on making a 3-view assumption and then using 3 separate pieces as experts and look for a consensus for this experts. This sort of ideas have been explored before: Estimating Accuracy from Unlabeled Data: A Bayesian Approach. ICML 2016 and references in that paper. It would make sense to compare with that line of work. - Another line of work that is related is "discrepancy"-based approaches of Ben-David et al. and Mohri et al. for domain adaptation. Note that these do not require stringent 3-view assumptions and give data-dependent guarantees (unlike results presented in this paper). - Can 3-view assumption be tested somehow? - How realistic is this assumptions? It seems that (at least for log reg) this is kind of equivalent to Naive Bayes, which almost never holds... - How robust are the result if 3-view assumption fails? Can we quantify this somehow? - Experiments are a little underwhelming. It would be nice to see more results than just one dataset. Also, I would to see robustness of this explored. In practice, it seems that we will never know that 3-view assumption holds, so it is crucial to see how robust this methodology is. - There is a whole section on different structures but no experiments with those so again it would be nice to see those results as well. Also experiments with different losses since this aspect of the paper is heavily emphasized.

Confidence in this Review

2-Confident (read it all; understood it all reasonably well)


Reviewer 6

Summary

The paper suggest the method of moments as a tool to estimate the error of a classifier in an unsupervised manner. The paper makes an assumption that the classifier can be decomposed into three conditionally independent parameters. This is indeed a strong generative assumption nevertheless the authors are able to prove a strong result. The authors show how this can be exploited in a semisupervised task inorder to exploit unlabeled data and few labeled examples to enhance performance.

Qualitative Assessment

The result of the papers are interesting and exploits interesting and non-trivial tools of method of moments and tensor decomposition. The exploitation of tensor decomposition within semi-supervised setting is interesting and does seem to lead to significant improvement. The assumptions made are indeed strong and it is not clear how one can verify it in practice. Nevertheless the paper does give an interesting method to perform classification, worthy of future research.

Confidence in this Review

2-Confident (read it all; understood it all reasonably well)